# Postnatal Dynamic Ciliary ARL13B and ADCY3 Localization in the Mouse Brain

**DOI:** 10.3390/cells13030259

**Published:** 2024-01-30

**Authors:** Katlyn K. Brewer, Kathryn M. Brewer, Tiffany T. Terry, Tamara Caspary, Christian Vaisse, Nicolas F. Berbari

**Affiliations:** 1Department of Biology, Indiana University-Indianapolis, 723 W. Michigan St., Indianapolis, IN 46202, USA; brewerka@iu.edu (K.K.B.); katmbrew@iu.edu (K.M.B.); 2Department of Human Genetics, Emory University School of Medicine, Atlanta, GA 30322, USA; tiffany.terry@emory.edu (T.T.T.); tcaspar@emory.edu (T.C.); 3Diabetes Center and Department of Medicine, University of California San Francisco, San Francisco, CA 92697, USA; christian.vaisse@ucsf.edu; 4Stark Neurosciences Research Institute, Indiana University-Indianapolis, Indianapolis, IN 46202, USA; 5Center for Diabetes and Metabolic Diseases, Indiana University School of Medicine, Indianapolis, IN 46202, USA

**Keywords:** cilia, hypothalamus, ARL13B, ADCY3, energy homeostasis, nucleus accumbens, brain

## Abstract

Primary cilia are hair-like structures found on nearly all mammalian cell types, including cells in the developing and adult brain. A diverse set of receptors and signaling proteins localize within cilia to regulate many physiological and developmental pathways, including the Hedgehog (Hh) pathway. Defects in cilia structure, protein localization, and function lead to genetic disorders called ciliopathies, which present with various clinical features that include several neurodevelopmental phenotypes and hyperphagia-associated obesity. Despite their dysfunction being implicated in several disease states, understanding their roles in central nervous system (CNS) development and signaling has proven challenging. We hypothesize that dynamic changes to ciliary protein composition contribute to this challenge and may reflect unrecognized diversity of CNS cilia. The proteins ARL13B and ADCY3 are established markers of cilia in the brain. ARL13B is a regulatory GTPase important for regulating cilia structure, protein trafficking, and Hh signaling, and ADCY3 is a ciliary adenylyl cyclase. Here, we examine the ciliary localization of ARL13B and ADCY3 in the perinatal and adult mouse brain. We define changes in the proportion of cilia enriched for ARL13B and ADCY3 depending on brain region and age. Furthermore, we identify distinct lengths of cilia within specific brain regions of male and female mice. ARL13B+ cilia become relatively rare with age in many brain regions, including the hypothalamic feeding centers, while ADCY3 becomes a prominent cilia marker in the mature adult brain. It is important to understand the endogenous localization patterns of these proteins throughout development and under different physiological conditions as these common cilia markers may be more dynamic than initially expected. Understanding regional- and developmental-associated cilia protein composition signatures and physiological condition cilia dynamic changes in the CNS may reveal the molecular mechanisms associated with the features commonly observed in ciliopathy models and ciliopathies, like obesity and diabetes.

## 1. Introduction

Primary cilia are microtubule-based structures found on most cell types in the body, including neurons and glia throughout the central nervous system (CNS). Protruding as a solitary structure, these cell appendages originate from the mother centriole or basal body and form a unique signaling compartment from that of the rest of the cell [1,2]. Regulated localization of specific proteins to cilia is one mechanism by which they establish their signaling organizing capacity. For example, the ciliary membrane contains specific G protein-coupled receptors (GPCRs) while appearing to exclude other closely related GPCRs [3,4]. In addition, specific GTPases, like ARL13B, and adenylyl cyclases, like ADCY3, have prominent roles at cilia [5,6,7,8]. Mutations altering cilia function and protein localization are associated with a diverse set of phenotypes in mammals, and the associated disorders are collectively called ciliopathies [3,4,5,6,7,8]. Thus, their distinct membrane composition diversifies cilia specialization for mediating cell–cell communication and regulating diverse physiological and developmental processes [1,2].

Dynamic localization of cilia proteins is one mechanism cells deploy to regulate cilia signaling. This is especially apparent during embryonic development and tissue patterning [9], as cells utilize the Hedgehog (Hh) signaling pathway to establish proper mammalian anatomy [10]. Hh signaling requires cilia. The Hh pathway components display dynamic ciliary localization in the presence or absence of Hh ligands [11,12,13]. Specifically, when the Hh receptor Patched is bound by ligand, it leaves the cilia compartment, and the pathway mediator Smoothened accumulates in the cilia, resulting in downstream signaling via Gli transcription factors [14]. These dynamic cilia functions are well recognized and understood in development; however, the roles for dynamic localization of cilia signaling proteins in adult tissue homeostasis remain unclear.

Nowhere is the paucity of knowledge around cilia function more evident than in cells throughout the developing perinatal and adult brain. Recent work has suggested that cilia in adult tissues can be quite static and as old as their cell [15], while other work has suggested that cilia in the brain may be more dynamic in their morphology, distribution, and signaling capacity [16,17]. Additionally, there are many challenges to studying cilia within the brain.

Perhaps the most challenging hurdle to understanding cilia in the brain lies in our ability to visualize them. The conventional markers of cilia in cells and other tissues rely on staining for stabilized forms of tubulin (e.g., acetylated α-tubulin staining). This approach fails in the brain as neurons and glia have elegant and elaborate processes that possess multiple forms of stable tubulin, the classic conventional marker for cilia immunolabeling acetylated α-tubulin, and other stabilized tubulins are not specific to cilia in the CNS [18]. Another challenge is the diversity of cell types that appear to have unique cilia characteristics within the CNS. For example, neurons typically possess a single primary cilium; however, a specific subset of neurons in the medial preoptic area, called GnRH neurons, have been reported to possess 2–4 cilia [19]. This cilia diversity is also evident in tanycytes, a special cell type lining the 3rd ventricular wall, and cells of the choroid plexus and ependyma where both motile and primary cilia are present [20,21,22]. Little is known about the potential diversity of cilia on cells deep inside the brain. Thus, the field of neuronal cilia has been limited, although several recent electron microscopy studies have begun to reveal the structural diversity and connectivity of CNS cilia [23,24,25]. However, staining and assessing the protein composition of the cilia membrane remains a significant challenge.

Here, we sought to evaluate cilia mouse brain regions where their dysfunction has been associated with altered behaviors (i.e., hypothalamic nuclei and hyperphagia) by immunolabeling for two well-characterized cilia membrane proteins, ARL13B and ADCY3, and the basal body protein FOP [18,26,27]. Our goal was to broadly assess the fundamental composition, distribution, and length of these cilia markers in the brain at different ages and under different physiological conditions associated with cilia function. Specifically, we observed that cilia in different brain regions at different ages appear to be diverse in their composition and morphology. Cilia also appear to be dynamic in their composition and length based upon physiological conditions. These data suggest that CNS cilia are not only structurally diverse but also diverse in their signaling capacity.

## 2. Materials and Methods

### 2.1. Animals

The mice were all housed under a standard 12 h light/dark cycle with ad libitum food and water unless stated differently in an experiment. All animal protocols and procedures were performed in accordance with the Institutional Animal Care and Use Committee (IACUC) at Indiana University-Purdue University Indianapolis. For developmental assessments, C57BL/6J (stock #000664) mice were perfused at 8 weeks, 5 weeks, 3 weeks, and postnatal day 0 (P0). The cilia and basal body epitope tagged alleles *ARL13B-mCherry/Centrin2-GFP*, respectively (STOCK Tg(CAG-Arl13b/mCherry1Kv) and Tg(CAG-EGFP/CETN2)3-4Jgg/KvandJ, strain #027967), and referred to as *Arl13b-mCherry;Centrin2-EGFP*), wildtype (C57BL/6J stock #000664) [28,29], and the ARL13b cilia exclusion mutant allele (*Arl13b^V358A^*) [30,31].

### 2.2. Genotyping

The mice, 3 weeks of age and older, used for developmental timepoints were visually assessed for sex. P0 pups were visually assessed for sex [32], and tail snips were taken at the time of collection and genotyped for *Sry* as described [33]. Ear punches were genotyped as previously described: *Arl13b-mCherry;Centrin2-GFP* [34,35] and ARL13B cilia exclusion mutant allele (Arl13b^V358A^) [30,31].

### 2.3. Diet and Feeding Manipulations

The mice were fed a standard chow diet consisting of 13% fat, 58% carbohydrate, and 28.5% protein caloric content (catalog no. 5001, LabDiet). For diet-induced obesity studies, the mice were given a high-fat diet (HFD) consisting of 60% fat, 20% carbohydrate, and 20% protein caloric content starting at 8 weeks of age (Research Diets Inc., catalog no. D12492) for 10 weeks [36]. Calorie restriction conditions consisted of 8-week-old mice that received a reduced amount of standard chow for 10 weeks. These mice received 20% less calories of standard chow, which was calculated weekly from ad libitum pair fed controls and were fed daily 1 h before the onset of the dark cycle as previously described.

### 2.4. Fixation and Preparation of Slide Sections

Samples were harvested at stated developmental timepoints and following specific feeding manipulations as listed and previously described in [36]. Briefly, the mice were anesthetized with 0.1 mL/10 g of body weight dose of 2.0% tribromoethanol (Sigma Aldrich, St. Louis, MO, USA) and perfused transcardially with PBS, followed by 4% paraformaldehyde (PFA) (catalog no. 15710, Electron Microscopy Sciences, Hatfield, PA, USA) [36]. Subsequently, the brains were isolated and postfixed in PFA for 4 h at 4 °C and then cryoprotected. After cryoprotection with 30% sucrose in PBS for 16–24 h at 4 °C, the brains were embedded in optimal cutting temperature compound (OCT) and cryosectioned at 15 µm directly onto slides for staining.

### 2.5. Immunofluorescence

Cryosections were washed twice with PBS for 5 min and then permeabilized and blocked in a PBS solution containing 1% BSA, 0.3% Triton X-100, 2% (*v*/*v*) donkey serum, and 0.02% sodium azide for 40 min at room temperature. The sections were incubated with primary antibodies in blocking solution overnight at 4 °C. The primary antibodies include anti-ARL13B (1:300 dilution; catalog no. ABIN1304543, antibodies-online Inc., Limerick, PA, USA), ADCY3 (1:1000 dilution; catalog no. CPCA-ACIII, EnCor, Gainesville, FL, USA), and FGFR1OP [FOP] (1:500 dilution; catalog no. 11343-1AP; Proteintech, Rosemont, IL, USA). The sections were then washed with PBS before incubating with secondary antibodies for 75 min at room temperature. The secondary antibodies include donkey conjugated Alexa Fluor 488, 546, and 647 (1:1000; Thermo Fisher Scientific, Waltham, MA, USA) against appropriate species according to the corresponding primary. All primary and secondary solutions were made in the blocking solution described above. The slides were then washed in PBS and counterstained with Hoechst nuclear stain (1:1000; catalog no. H3570, Thermo Fisher Scientific) for 5 min at room temperature. Coverslips were mounted using SlowFade Diamond Antifade Mountant (catalog no. S36972, Thermo Fisher Scientific).

### 2.6. RNAScope In Situ Hybridization

Tissue sections (15 μm) were collected using our fixation and preparation protocol above and then prepped and pretreated with 4% PFA for 16 h at 4 °C, followed by part 2 of protocol TN 320534 (Advanced Cell Diagnostics (ACD), Newark, CA, USA) as described [37]. Following tissue preparation, the detection of transcripts was performed using an RNAscope 2.5 HD Duplex Detection Kit (Chromogenic) User Manual Part 2 (ACDBio Document 322500-USM, Newark, CA, USA). The slides were assayed using a probe specific to *Arl13b* (catalog no. 1044271-C2) transcripts (ACD), counterstained with hematoxylin, dehydrated, and mounted using VectaMount (Vectorlabs, Burlingame, CA, USA). Slides with a positive control probe (*PPIBC1*/*POLR2A*; catalog no. 321651) and negative control probe (*DapB*; catalog no. 320751) were run with each experiment. At least 3 animals were analyzed for each group.

### 2.7. Imaging

All immunofluorescent images were acquired using a Leica SP8 confocal microscope in resonant scanning mode using 63×. All images were collected with a bit depth of 16 and a zoom factor of 1.25 and captured as 1024 × 1024-pixel images. All colorimetric images of RNAscope were acquired using a Nikon 90i microscope with a color camera and Nikon Elements BR 4.13.05 software.

### 2.8. Image Data Analysis

Computer-assisted cilia analysis was performed as previously described [36,38]. Briefly, images encompassing z-stacks of 40 optical slices were sum projected and analyzed using the Nikon artificial intelligence 5.30.06 software module, which we have trained to recognize immunofluorescent stained and fluorescent reporter allele cilia. Cilia image datasets with ARL13B, ADCY3, and the basal body proteins Centrin2-GFP and FOP were used for training the software module. These datasets achieved a training loss of 0.011. As part of our approach, objects >1 μm in length and possessing a basal marker signal were included in the analysis. Six mice (three males, three females) were analyzed per experimental developmental condition, and 5 male mice were analyzed per experimental physiological condition with three images captured per specific brain region unless specified differently in an experiment.

### 2.9. Statistical Analysis

All statistical tests were performed using GraphPad Prism version 10.1.2. Specific statistical tests are described in each figure legend, and all statistically significant datasets are noted in the figures.

## 3. Results

### 3.1. Approaches for Cilia Visualization and Large-Scale Analysis in the Brain

While the brain is highly ciliated, we still do not understand cilia diversity or dynamics over developmental time, between sexes, across brain regions, and under different physiological conditions. We tested the hypothesis that cilia are fundamentally different in their morphology (length) and composition (protein localization) across these parameters in the brain. To evaluate cilia in the mouse brain, we modified our previously published computer-assisted, image-analysis approach for evaluating large amounts of cilia and basal bodies in confocal images (Figure 1 and Figure 2A) [36,38]. Initially, we used a well-characterized ARL13B transgenic fusion fluorescent allele, *Arl13b-mCherry*, in conjunction with a fluorescent basal body allele, *Centrin2-EGFP*, here after referred to as *Arl13b-mCherry;Centrin2-GFP* (Figure 2A, “Visualization: Fluorescent Alleles”) [28]. Co-stained with an antibody for the neuronal cilia marker adenylate cyclase III (ADCY3), our qualitative assessment of these regions revealed three distinct cilia membrane compositions (Figure 2B). These included cilia enriched with only ARL13B (red cilia, hereafter ARL13B+), cilia enriched with only ADCY3 (green cilia, hereafter ADCY3+), and cilia localizing both ARL13B and ADCY3 (orange cilia, hereafter COLO) (Figure 2A,B) [28,36,38]. We chose to focus on several brain regions where cilia have been implicated in behaviors such as feeding and reward [39,40,41]. These brain regions include the hypothalamic nuclei, such as the arcuate nucleus (ARC), paraventricular nucleus (PVN), ventromedial hypothalamus (VMH), and suprachiasmatic nucleus (SCN), as well as the shell (NAs) and core (NAc) of the nucleus accumbens (Figure 2C).

Our analysis further distinguishes the regions of localization within individual cilia. For example, identified COLO cilia are assessed in two ways, the length of ARL13B localized within COLO cilia (denoted as ARL13B in COLO) and the length of ADCY3 localized within COLO cilia (denoted as ADCY3 in COLO), which allows us to assess specific protein localization changes within the population of COLO cilia by recognizing that ARL13B and ADCY3 do not completely overlap within any given cilium (Figure 2). We observed significantly longer ADCY3+ cilia as well as ADCY3 length in COLO cilia in the *Arl13b-mCherry;Centrin2-EGFP* samples in several brain regions, including the arcuate nucleus (ARC), paraventricular nucleus (PVN), and the core and shell of the nucleus accumbens (NAc and NAs) (Figure 2D). Similar to the findings of length changes previously observed in transgenic ARL13B-GFP fusion allele mice [6], these results suggest that the transgenes used to assess ARL13B might alter cilia lengths. As predicted for a transgenic allele, RNAscope in situ hybridization for the *Arl13b* transcript revealed overt increases in *Arl13b* mRNA labeling in the *Arl13b-mCherry;Centrin2-EGFP* transgenic model compared to the C57BL/6J control samples (Figure 2E). Immunostaining for both ARL13B-mCherry and ARL13B protein in *Arl13b-mCherry;Centrin2-EGFP* mice showed a near complete colocalization of ARL13B and mCherry (Figure 3A,B). Additionally, in another control experiment for antibody specificity, the ARL13B immunofluorescence staining failed to mark ARL13B+ and COLO cilia in brain samples from mice carrying a missense mutation in ARL13B at amino acid position 358 (*Arl13b^V358A^*), which excludes ARL13B from localizing to cilia [30,31] (Figure 3C,D). Therefore, we pursued immunofluorescence approaches in the C57BL/6J wildtype mice to assess cilia in brain regions associated with ciliopathy phenotypes and dynamic cilia changes, such as feeding centers of the hypothalamus and the SCN [17,36,42,43]. These results led us to evaluate ARL13B and ADCY3 through co-immunofluorescence in different brain regions, sexes, ages, and feeding conditions.

### 3.2. Neuroanatomical Region Dependent Cilia Signatures in the Adult and Perinatal (P0) Brain

The staining of several nuclei and brain regions in the adult (Figure 4A) revealed cilia distribution signatures unique to these different regions (Figure 4B). For example, a majority of cilia in the ARC are enriched with only ADCY3 (ADCY3+), while the majority of cilia in the nucleus accumbens localize both ARL13B and ADCY3 (COLO) (Figure 4B). While our initial analysis consisted of 8-week-old adult animals, ARL13B plays critical roles in embryonic neurodevelopment. Therefore, we also examined ciliary ARL13B and ADCY3 in postnatal day 0 (P0) brains, comparing the same regions in males and females (Figure 5A). A direct comparison between P0 male and female pups did not show differences in localization of ADCY3 or ARL13B, or colocalization of the two proteins (Figure 5B); however, it did reveal that the nuclei involved in feeding and the accumbens possess a majority of ARL13B+ cilia, whereas the SCN possesses 25% of ADCY3+ and COLO cilia (Figure 5B). Additionally, at P0, ADCY3 cilia staining in the brain was relatively rare compared with ARL13B, which appeared to be the more prominent protein to localize to the ciliary membrane (Figure 5).

### 3.3. Sex-Dependent Cilia Length Differences

Interestingly, we observed sexually dimorphic cilia lengths in animals that were 3 WKS and younger (Figure 6A,B). For example, at P0, ARL13B+ cilia were longer in the PVN and SCN of females (Figure 6C). In addition, in 3-week-old female mice, ARL13B in COLO cilia of the accumbens shell and ADCY3 in COLO cilia of the PVN were significantly longer than males (Figure 6D). However, the lengths remained largely unchanged between sexes at older ages (5 WKS and 8 WKS) (Figure 7). We also noted that ARL13B+ cilia were generally shorter than ADCY3+ cilia at 3 WKS and older. In addition, COLO cilia had lengths consistent with ADCY3+ alone cilia (Figure 6 and Figure 7). Representative images of the ARC are provided, which is a region implicated in ciliopathy-associated obesity.

### 3.4. Cilia Signatures in the Brain at Different Postnatal Developmental Ages

The stark difference in cilia protein enrichment between the P0 and adult CNS led us to examine intervening timepoints to learn if there is an age when adult CNS cilia signatures are established. We examined brains at 3 weeks of age (3 WKS, weaning), 5 weeks of age (5 WKS, prior to sexual maturation), and 8 weeks of age (Figure 8). Here, we observe and provide images of the nucleus accumbens as it appears more dynamic throughout postnatal development with ARL13B+, ADCY3+, and COLO cilia populations changing between 3 WKS, 5 WKS, and 8 WKS in the core and shell (Figure 8A and Figure 8B, respectively). The nuclei of the hypothalamus like the ARC, SCN, and PVN; however, after 3 WKS, cilia signatures consist predominantly of ADCY3+ cilia (Figure 8B).

We also evaluated cilia lengths within the same regions across all ages and found ARL13B+ cilia do not significantly alter their length at any time point (Figure 9A), whereas ARL13B length in COLO cilia is significantly longer in the PVN and accumbens between P0 and 3 weeks of age and shorter in the SCN between 3 and 5 weeks of age (Figure 9B). Interestingly, we did observe a trend of shorter ARL13B length in COLO cilia at 5 WKS in the male PVN and female SCN (Figure 9B). ADCY3+ cilia and ADCY3 in COLO cilia become significantly longer as animals become older (Figure 10A,B).

### 3.5. Impact of Energy Homeostasis Changes on Cilia Lengths in the Hypothalamus

In our previous studies, we have observed changes in cilia GPCR composition based on physiological conditions [36]. Furthermore, our data have indicated the requirement for ciliary ARL13B in energy homeostasis [31]. In the current study, we sought to determine the impact of physiological conditions associated with hypothalamic functions like feeding on the lengths and distributions of ARL13+, ADCY3+, and COLO cilia. We established cohorts of mice that were ad libitum fed, calorically restricted, and obese on a high-fat diet (HFD). These mice were maintained under these feeding conditions for 11 weeks and weighed weekly (Figure 11A). Characterizing the cilia under these conditions revealed no significant changes in length in ARL13B+ or COLO cilia (Figure 11B,C). However, ADCY3+ cilia were significantly longer in the PVN under caloric restrictions and HFD obese conditions (Figure 11C). No significant differences in the protein composition of the cilia membrane were observed. Although, we did observe a trend toward protein localization to cilia changes within the PVN with less COLO cilia in HFD obese animals (Figure 11D).

## 4. Discussion

Cilia in the CNS are known to be structurally diverse and vary with their signaling capacity, but understanding cilia on various nuclei of the brain, at different ages, and under physiological conditions, and understanding how cilia become established in the mature brain are relatively unknown. Herein, we used our previously developed computer-assisted approach to characterize cilia in large numbers across several brain regions where cilia dysfunction is associated with behavioral changes (i.e., hypothalamic cilia and circadian rhythms and feeding behaviors) [44,45,46,47].

Initially, we assessed cilia using transgenic alleles for cilia membrane-associated ARL13B (Arl13b-mCherry) and the basal body (Centrin2-EGFP) with the idea that fluorescent alleles may provide a more-efficient, less-biased, and uniform approach. Similar to what is observed in the hippocampus [48], our analysis revealed cilia with the ARL13B-mCherry transgene were longer in the hypothalamus and accumbens, suggesting that cilia visualization alleles themselves may impact cilia morphology and function (Figure 2). We, therefore, optimized an antibody-based approach and found that co-staining for ARL13B and ADCY3 with the basal body marker FOP identified cilia broadly; the loss of ARL13B staining in the cilia exclusion allele context (*Arl13b^V358A^* homozygotes) also built confidence in this antibody-reliant approach (Figure 3). One potential caveat to our technique would be antibody epitope masking that could occur within the cilia under certain conditions. However, given the strong correlation in the double labeling and lack of staining in the cilia excluded allele, *Arl13b^V358A^*, epitope masking is a minor concern at best.

Cilia are known to have different characteristics in different brain regions, including length, receptor localization, or loss of the organelle upon maturation [49]. Most work has shown changes in the localization of specific, signaling receptors, most commonly GPCRs (e.g., odorant receptors [50], 5HT6 [51], MCHR1, and NPY2R [36,52]), to regulate their functions and signaling capacities [53]. Additionally, many of these investigated receptors have implicated roles in specific brain nuclei and physiological processes, suggesting that their dynamic properties may only be observed at and under precise conditions [36]. One challenge in the field is understanding if more ubiquitous signaling proteins, like proteins that produce and respond to secondary messengers, also have specific dynamics to regulate their functions and signaling capacities at cilia.

We characterized cilia more broadly by choosing two cilia membrane proteins, ARL13B and ADCY3. ARL13B is well-characterized as ciliary in many cellular contexts and in neural development [54], and ADCY3 was initially identified in olfactory cilia and remains the most broadly expressed neuronal cilia marker [55,56]. Using these two cilia proteins, we were able to identify multiple cilia with different membrane make-ups and then assess region-specific cilia signatures based on these protein compositions.

Initially, we were surprised to observe that most cilia in certain regions of the adult brain did not appear to possess ARL13B (Figure 4: ARC). ARL13B is thought to be a very prominent cilia marker in most mammalian cell systems and tissues. The prevalence of ciliary ARL13B in embryonic tissues, including the developing neural tube and brain, is well-characterized [6]. We sought to evaluate the brain of younger animals for ciliary ARL13B. On the day of birth (P0), we observed very prominent levels of ciliary ARL13B. In contrast, ADCY3 cilia at this age were relatively rare (Figure 5). These data suggested that cilia undergo a maturation process in the brain in which they are initially associated with ARL13B, and as animals age, ADCY3 becomes the predominant cilia marker in the brain. Future studies will determine the protein signatures and the potential signaling significance on cilia maturation and the observed transition from ARL13B to ADCY3 cilia in the CNS.

To determine when adult cilia signatures become established, we assessed cilia in the brains of mice at two additional ages between P0 and 8-week-old adults. We chose 3 weeks of age as this is a weaning period and associated with changes in hypothalamic-mediated feeding behaviors and diet [57]. We also chose 5 weeks of age as this is closer to adulthood, after feeding behaviors are established, and prior to sexual maturation [58]. We were surprised to observe that cilia signatures appear to be established throughout the hypothalamus by 3 weeks, whereas the adult accumbens signatures remain dynamic at each age that we quantified (Figure 8). These observations provide additional support for the hypothesis that cilia in the brain undergo a maturation process (as observed with changes in ciliary protein enrichment), and these changes in cilia composition also correlate with the maturation of individual brain regions. For example, the hypothalamus is largely established at 3 weeks [59], while the accumbens continues to mature in rodents through adolescence and into adulthood [60,61,62,63]. Alternatively, individual brain regions, like the nucleus accumbens and hypothalamic nuclei, may display unique cilia composition dynamics as part of their homeostatic functions in the mature brain. Future studies will seek to understand if cilia signatures change with behavior and electrophysiological activities associated with specific brain nuclei.

In addition to age and anatomical region-specific cilia signatures, we also observed sex-dependent differences on the day of birth (P0) in the length of ARL13B+ hypothalamic cilia. These findings at such an early age suggest that ARL13B may play differential roles in males and females during perinatal development or that ARL13B ciliary localization could be influenced by embryonic hormone secretion [64]. Similarly, the drastic change in cilia composition throughout development, from mostly ARL13B+ in P0 animals to primarily ADCY3+ cilia at 3 WKS and older, could be caused by altered cell populations and signaling that are associated with major life changes, such as changes in feeding [65], social environment [66,67], and sexual maturation [68]. Thus, these data suggest that both embryonic and adult environmental changes that alter brain architecture could be correlated with the protein make-up of cilia.

We recently demonstrated that GPCR localization to cilia of the hypothalamus can be dynamic under different feeding and body composition conditions. Here, we assessed cilia under the same feeding parameters, including HFD-induced obesity and caloric restriction. We did not observe robust cilia length or composition changes in either feeding paradigm except in the PVN where ADCY3+ cilia lengths are longer in calorically restricted and HFD obese mice compared to ad libitum controls (Figure 11). Based on these and our previously published work, dynamic cilia properties may be protein (e.g., NPY2R vs. ADCY3) and region (e.g., ARC vs. PVN) specific [36]. These data continue to implicate that dynamic changes in cilia morphology can be associated with the known function of the brain region (e.g., feeding behavior and the PVN).

## 5. Conclusions

Together, these observations and data continue to add to the complexity of cilia dynamics and potential interactions among specific mouse brain regions. Future studies will need to assess the potential differences between mouse and human CNS cilia. Here, we show that cilia in the mouse brain appear dynamic throughout early postnatal development, as cilia protein composition and distribution change in the hypothalamus and accumbens. Specifically, ARL13B localizes primarily to cilia during early development, and as the animals age, ARL13B is lost from the cilium, and ADCY3 becomes a major component of the cilia membrane. This suggests that ARL13B may be critical for establishing cilia membrane properties early in postnatal development and ADCY3 is required for signaling of cells once their cellular identities and/or circuitry are fully established. Thus, future studies can now better explore cilia dynamics in specific tissues and cell types to understand composition and function. Our observations also raise several big-picture questions about cilia in the brain that will be the basis of future work: what cell types and time scales CNS cilia use and how they influence behaviors.

## Figures and Tables

**Figure 1 cells-13-00259-f001:**
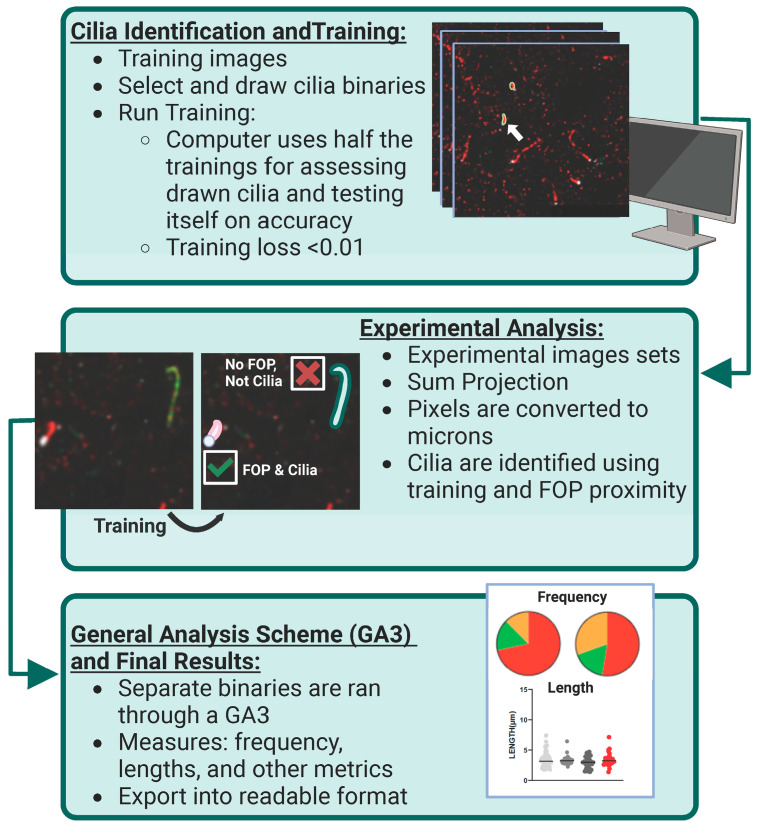
AI identification and analysis of ciliary ARL13B and ADCY3. Cilia identification and training: To identify and assess cilia localization of ARL13B and ADCY3 in a robust, non-biased manner, Nikon Elements Analysis (NIS Elements) was used to recognize proteins on sample images. Hand-drawn binaries were created on the sample set of data and then ran overnight using Segment.ai. Here, the computer uses half of the training set to teach itself the characteristics of the identified cilia and the other half to test itself on accuracy. The training loss calculated at the end of our run was found to be 0.01. White arrow indicates an example of a selected cilia binary. Experimental analysis: Once the training set is established (goal of training loss to be <0.05), experimental image sets can be run through NIS Elements using this trained Segment.ai to identify both ARL13B and ADCY3 cilia. Experimental z-stacks are made into sum projection images and converted to microns based on image capture settings. Separate cilia binaries are identified for each cilia marker on their respective channels. To enhance accuracy, only cilia binaries in contact with an FOP basal body marker are selected for final analysis. General analysis scheme (GA3) and final results: Using the finalized cilia binaries, a GA3 is constructed to measure specific cilia characteristics, such as frequency and length. Separate GA3s were constructed to assess the cilia characteristics of single binaries and overlapping cilia binaries. All data are exported and assessed into Excel and GraphPad Prism for further analysis.

**Figure 2 cells-13-00259-f002:**
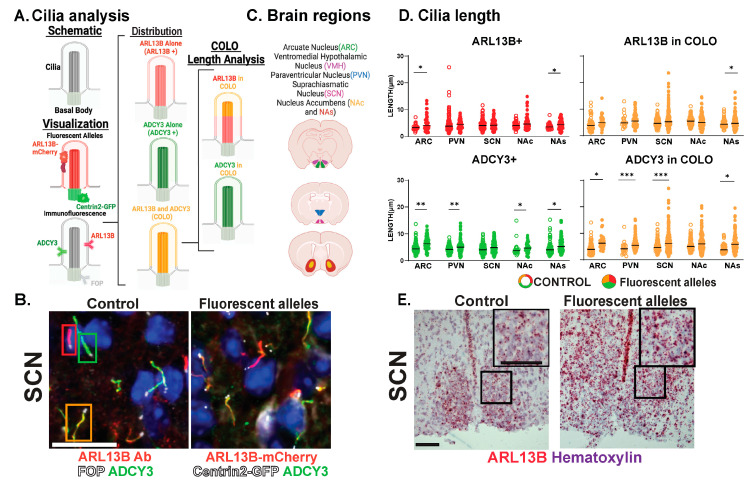
Assessing CNS cilia with an ARL13B fluorescent allele. (**A**) Schematic: Cilia and basal body. Visualization: Fluorescent transgenic alleles. Their localization to cilia membrane-associated ARL13B-mCherry and basal body-associated Centrin2-GFP. Immunofluorescence. Proteins ADCY3 and ARL13B enriched along the cilia membrane and FOP to the basal body analyzed throughout. Distribution: Cilia with ARL13B alone (ARL13B+), with ADCY3 alone (ADCY3+), or with both ARL13B and ADCY3 (COLO). Cilia lengths assessed in two ways for COLO cilia. ARL13B measured length (ARL13B in COLO) or ADCY3 measured length (ADCY3 in COLO). (**B**) Immunofluorescence for ARL13B in the suprachiasmatic nucleus (SCN) of 8-week-old C57BL/6J control and *Arl13b-mCherry;Centrin2-EGFP* mice. Cilia markers are indicated with ADCY3 (green), ARL13B antibody labeled cilia (red), and basal body marker FOP (white) in C57BL/6J control animals and *Arl13b-mcherry* and *Centrin2-GFP* animals. Boxes indicate examples of cilia with different membrane compositions: ARL13B+ (red), ADCY3+ (green), and COLO (orange) boxes, respectively. Scale bar 10 µm. Hoechst-stained nuclei blue. (**C**) Brain regions assessed. (**D**) Analysis of cilia lengths between C57BL/6J control (open circles) and *Arl13b-mCherry;Centrin2-EGFP* (closed circles) brain regions. Significant length differences are indicated * *p* < 0.05, ** *p* < 0.01, *** *p* < 0.001, using nested *t*-test. N = 10 animals (6 C57BL/6J control and 4 *Arl13b-mCherry;Centrin2-EGFP*). (**E**) Colorimetric RNAscope in situ hybridization in the SCN of 8-week-old animals for *Arl13b* (red) in C57Bl/6J (control) and *Arl13b-mCherry;Centrin2-EGFP* animals. Black boxes indicate region for inset images. Counter-stain hematoxylin (purple). Scale bar 60 µm. N = 4 animals (2 C57BL/6J control and 2 *Arl13b-mCherry;Centrin2-EGFP*). All animals are 8-week-old adults.

**Figure 3 cells-13-00259-f003:**
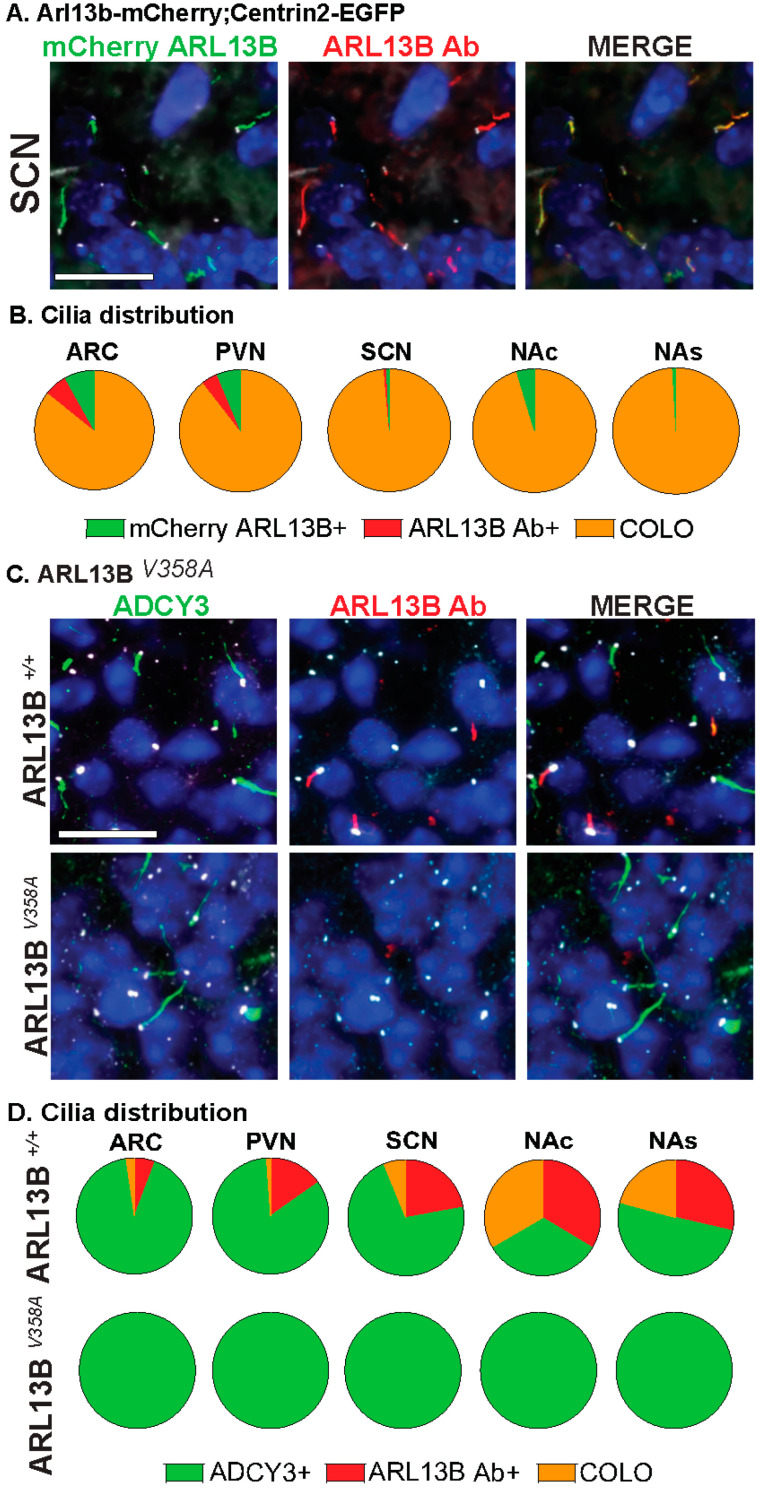
Assessing CNS cilia with an antibody approach. (**A**) Immunofluorescence for mCherry (green) and ARL13B (red) in the suprachiasmatic nucleus (SCN) of adult *Arl13b-mCherry;Centrin2-EGFP* mice. Centrin2-EGFP basal body marker in white. (**B**) Cilia distribution in the ARC, PVN, SCN, NAc, and NAs of *Arl13b-mCherry;Centrin2-EGFP* animals with mCherry cilia (mCherry ARL13B+, green) and ARL13B antibody labeled (ARL13B Ab+, red) and cilia localizing both (COLO). N = 4 to 8 *Arl13b-mCherry;Centrin2-EGFP* animals per region. (**C**) Immunofluorescence for ARL13B (red) and cilia marker ADCY3 (green) in the SCN of mice homozygous for the cilia restricted *Arl13b* allele (*Arl13b^V358A^*). Basal body staining for FOP in white. (**A**,**C**) Scale bars 10 µm. Hoechst-stained nuclei blue. (**D**) Cilia distribution analysis shows complete absence of ARL13B positive cilia and no changes in ADCY3 positive cilia in *Arl13b^V358A^* homozygous animals in the hypothalamus (ARC, PVN, SCN) and accumbens (NAc, NAs). N = 4 animals (2 *Arl13b^+/+^* and 2 *Arl13b^V358A^*). All animals are 8-week-old adults.

**Figure 4 cells-13-00259-f004:**
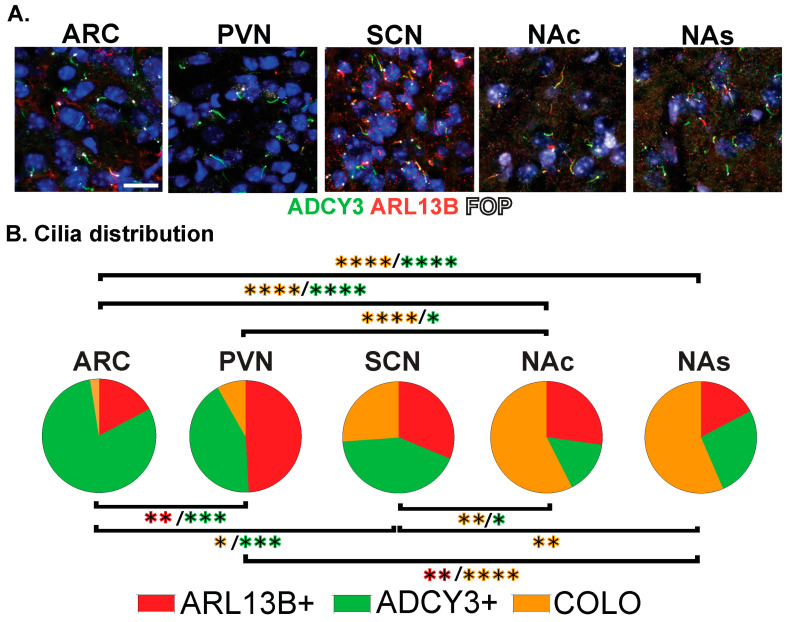
Region-dependent cilia signatures in the adult mouse brain. (**A**) Immunofluorescence for ADCY3 (green), ARL13B (red), and basal body marker FOP (white) in the arcuate nucleus (ARC), paraventricular nucleus (PVN), and suprachiasmatic nucleus (SCN) of the hypothalamus and nucleus accumbens core (NAc) and shell (NAs). Scale bar 10 µm. Hoechst-stained nuclei blue. (**B**) Cilia distribution analysis of membrane composition: ARL13B+, ADCY3+, and COLO. Two-way ANOVA and Tukey’s multiple comparisons analyses revealed differences among all regions analyzed, ARC, PVN, SCN, NAc, and NAs, in 8-week-old mice. Significant differences are indicated * *p* < 0.05, ** *p* < 0.01, *** *p* < 0.001, **** *p* < 0.0001. N = 6 animals (3 males and 3 females). All animals are 8-week-old adults.

**Figure 5 cells-13-00259-f005:**
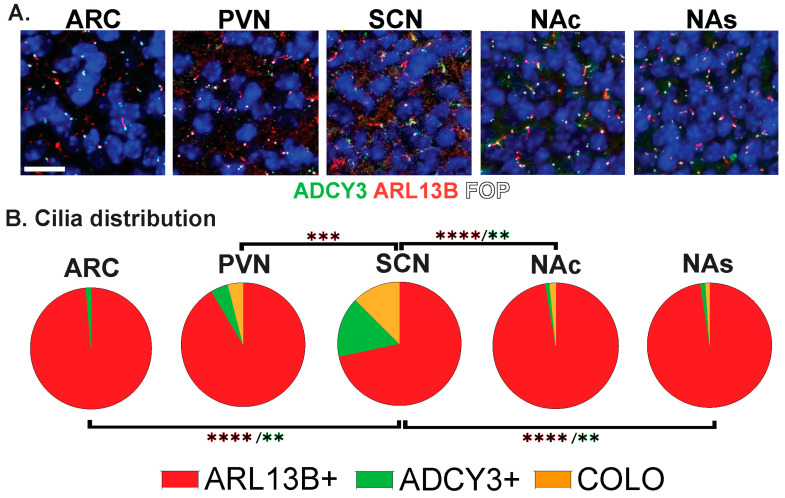
Region-dependent cilia signatures in perinatal mouse brain. (**A**) Immunofluorescence for ADCY3 (green), ARL13B (red), and basal body marker FOP (white) in the arcuate nucleus (ARC), paraventricular nucleus (PVN), and suprachiasmatic nucleus (SCN) of the hypothalamus and nucleus accumbens core (NAc) and shell (NAs) on the day of birth (P0) in males and females. Scale bar 10 µm. Hoechst-stained nuclei blue. (**B**) Cilia distribution analysis of cilia membrane composition: ARL13B+, ADCY3+, and COLO. Two-way ANOVA and Tukey’s multiple comparisons analyses revealed differences among specific regions in P0 mice. Significant differences are indicated ** *p* < 0.01, *** *p* < 0.001, **** *p* < 0.0001. N = 6 animals (3 males and 3 females). All animals are postnatal day 0 (P0).

**Figure 6 cells-13-00259-f006:**
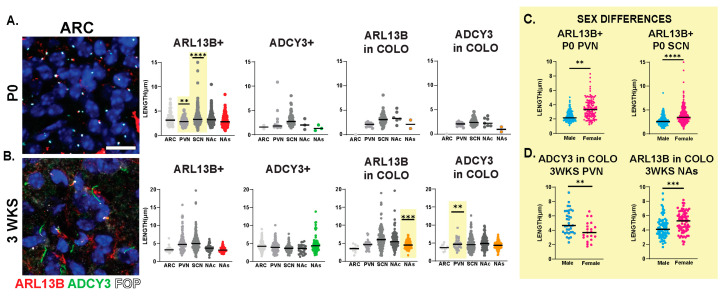
Sex differences in cilia length in early postnatal development. Analysis of cilia length across regions at different ages. (**A**) Immunofluorescence and cilia length analysis for ADCY3 (green), ARL13B (red), and basal body marker FOP (white) in the arcuate nucleus (ARC) at the day of birth (P0). Scale bar 10 µm. Hoechst-stained nuclei blue. (**B**) Statistically significant differences in cilia lengths between males and females in the ARL13+ cilia lengths in the P0 PVN and SCN. (**C**) Immunofluorescence and cilia length analysis for ADCY3 (green), ARL13B (red), and basal body marker FOP (white) in the arcuate nucleus (ARC) at 3 weeks of age (3 WKS). Scale bar 10 µm. Hoechst-stained nuclei blue. (**D**) ARL13B in COLO cilia length in 3 WKS NAs and ADCY3 in COLO cilia length in 3 WKS PVN. Nested *t*-test analysis revealed significant differences, which are indicated ** *p*< 0.01, *** *p*< 0.001, **** *p*< 0.0001. N = 6 animals (3 males and 3 females).

**Figure 7 cells-13-00259-f007:**
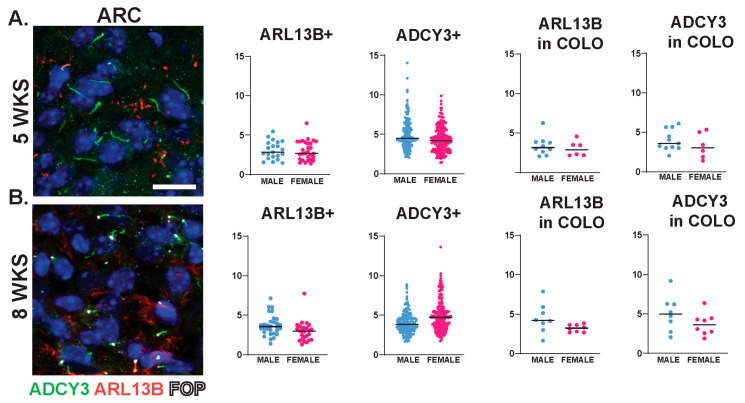
Cilia length is similar between sexes across neuroanatomical regions past 5 WKS of age. (**A**,**B**) Analysis of cilia length across regions at 5 and 8 WKS. Immunofluorescence and cilia length analysis for ADCY3 (green), ARL13B (red), and basal body marker FOP (white) in the arcuate nucleus (ARC) of 5- and 8-week-old animals (5 WKS, 8 WKS). Scale bar 10 µm. Hoechst-stained nuclei blue. Scale bar 10 µm. Nested *t*-test analysis did not reveal differences between males and females. N = 6 animals (3 males and 3 females).

**Figure 8 cells-13-00259-f008:**
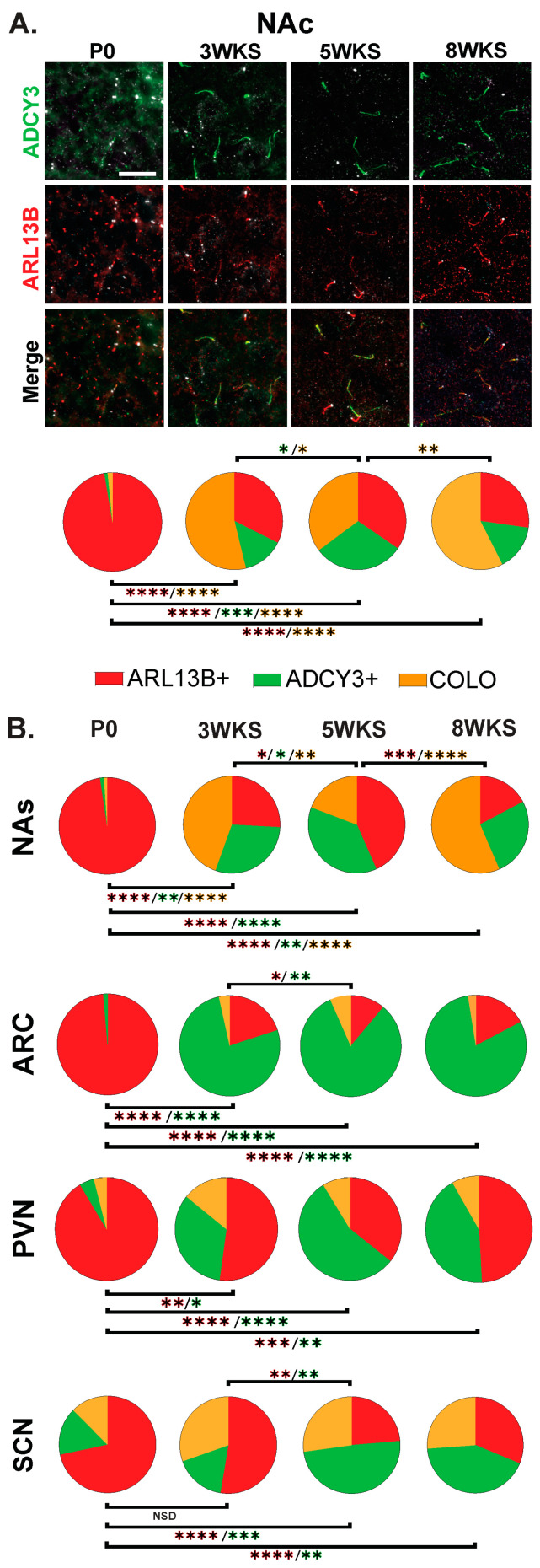
Age-dependent cilia signatures in the mouse brain. (**A**) Immunofluorescence and cilia distribution analysis for ADCY3 (green), ARL13B (red), and basal body marker FOP (white) in the nucleus accumbens core (NAc) at the day of birth, 3 weeks, 5 weeks, and 8 weeks of age (P0, 3 WKS, 5 WKS, and 8 WKS). Scale bar 10 µm. Hoechst-stained nuclei blue. (**B**) Cilia distribution analysis of ciliary protein enrichment: ARL13B+, ADCY3+, and COLO. Two-way ANOVA and Tukey’s multiple comparisons analyses revealed age differences within a region. Significant differences are indicated * *p* < 0.05, ** *p* < 0.01, *** *p* < 0.001, **** *p* < 0.0001. N = 6 animals per age (3 males and 3 females). Male and female data were combined for this analysis.

**Figure 9 cells-13-00259-f009:**
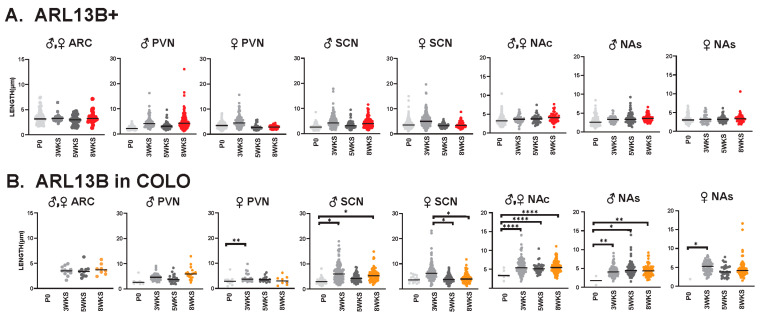
ARL13B ciliary lengths are similar throughout postnatal development. Analysis of ARL13B cilia lengths in each brain region (ARC, PVN, SCN, NAc and NAs) at the day of birth, 3 weeks, 5 weeks, and 8 weeks of age (P0, 3 WKS, 5 WKS, and 8 WKS). No significant differences were observed between male (**♂**) and female (♀) animals in the ARC and NAc. Datasets were combined when no sex-dependent differences were observed. (**A**) Cilia with only ARL13B (ARL13B+) showed no difference. (**B**) Cilia length of the ARL13B mask in cilia with both ARL13B and ADCY3 (ARL13B in COLO). Two-way ANOVA and Tukey’s multiple comparisons analysis revealed significant differences, which are indicated * *p* < 0.05, ** *p* < 0.01, **** *p* < 0.0001. N = 6 (3 males and 3 females).

**Figure 10 cells-13-00259-f010:**
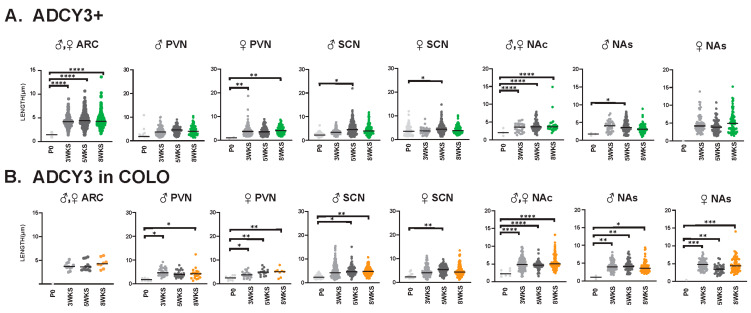
ADCY3 ciliary lengths are significantly longer after 3 WKS of age in different neuroanatomical regions. Analysis of ADCY3 cilia lengths in each brain region (ARC, PVN, SCN, NAc and NAs) at different ages (P0, 3 WKS, 5 WKS, and 8 WKS). Significant differences between male and female cilia lengths are indicated in separate graphs for the PVN, SCN, and NAs (**♂** male, ♀ female). Datasets were combined when no sex-dependent differences were observed. (**A**) Cilia with only ADCY3 (ADCY3+). (**B**) Cilia length of the ADCY3 mask in cilia with both ARL13B and ADCY3 (ADCY3 in COLO). Two-way ANOVA and Tukey’s multiple comparisons analyses revealed significant differences, which are indicated * *p* < 0.05, ** *p* < 0.01, *** *p* < 0.001, **** *p* < 0.0001. N = 6 (3 males and 3 females).

**Figure 11 cells-13-00259-f011:**
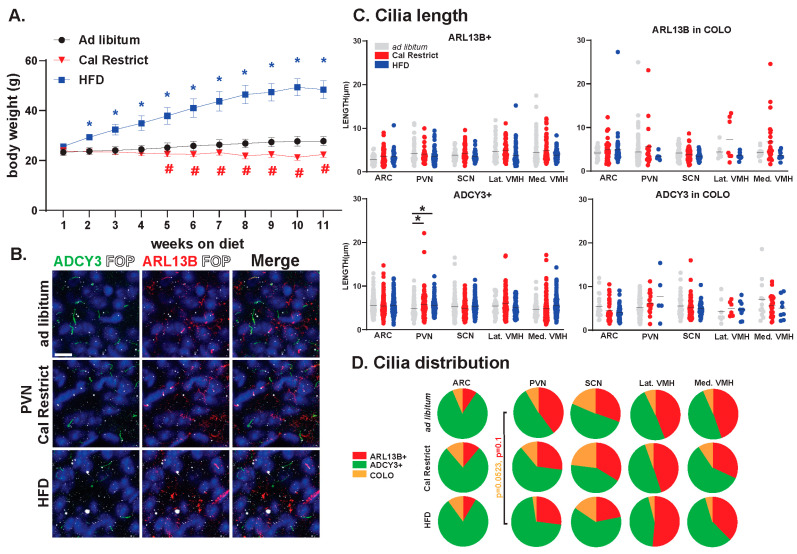
Physiological condition-dependent cilia analysis in the mouse brain. Analysis of cilia under different feeding and body composition conditions, ad libitum fed, high-fat diet-induced obesity (HFD) and pair-fed caloric restricted (Cal Restrict). (**A**) Weekly body weight measurements in grams (g) beginning at 8 weeks of age. ANOVA analyses revealed significant differences, which are indicated with blue * and red # when * *p* < 0.05, (**B**) Immunofluorescence for ADCY3 (green), ARL13B (red), and basal body marker FOP (white) in the paraventricular nucleus (PVN) for each condition. Scale bar 10 µm. Hoechst-stained nuclei blue. (**C**) Analysis of cilia lengths between conditions. Significant length differences are indicated * *p* < 0.05 using two-way ANOVA and Tukey’s multiple comparisons. (**D**) Analysis of cilia distribution: ARL13B+, ADCY3+, and COLO. Two-way ANOVA and Tukey’s multiple comparisons analyses revealed no significant differences among specific regions in ad libitum, Cal Restrict, or HFD. Trends in PVN cilia distribution between ad libitum and HFD are indicated (*p* = 0.0523 for COLO and *p* = 0.1 for ARL13B+). N = 4 male animals per physiological condition (4 ad libitum, 4 Cal Restrict, and 4 HFD). All mice went on diet or caloric restriction at 8 weeks of age and were analyzed at 19 weeks of age.

## Data Availability

The raw data supporting the conclusions of this article will be made available by the authors on request.

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
