# Peer review of "Postnatal Dynamic Ciliary ARL13B and ADCY3 Localization in the Mouse Brain"

_cells, 2024, doi:10.3390/cells13030259_

Round 1
Reviewer 1 Report
Comments and Suggestions for Authors
This manuscript by Brewer et al. of the Berbari-group on the dynamics of cilia in the mouse brain is interesting and important. The study is devised by leading experts in the field of cilia biology and the experiments are conducted in an elegant and well-controlled manner. An important point is made at the beginning, namely, an antibody-based immunostaining approach proved more reliable to detect primary cilia in mouse brain than using transgenic animals in which visualization alleles (mCherry tagged ARL13B and GFP-tagged centrin) were expressed and caused alterations in cilia structure. This is a crucial experimental constrain to be considered also in future studies on cilia dynamics in the central nervous system. The results achieved with brain cryosections from C57BL6 mice of different ages and sexes demonstrate that the ciliary membrane protein ARL13B is predominantly detectable on primary cilia in various brain regions of perinatal and young mice, while adenyl cyclase ADCY3 is a lot more prominent on cilia at older ages of the animals. Interestingly, the authors propose that different functions are played by either ciliary component, namely, in ciliogenesis during embryonal and early post-natal neurodevelopment (ARL13B), while ADCY3 seems more important in establishing functional features of primary cilia at adult states. Sexual dimorphism was observed in the lengths of primary cilia in mice of up to 3 weeks of age, whereas older animals did not exhibit cilia differences in males and females. Furthermore, the primary cilia composition and lengths are characterized in different mouse brain regions, thus, revealing that the primary cilia of cells in the paraventricular nucleus (PVN) are more dynamic in length and composition than those of other brain regions. This aspect was further tested and cilia dynamics were correlated in animals fed with different diets that indeed resulted in length and ADCY3-selective composition changes exclusively in the PVN.
There are few remarks that need attention:
1. Abstract, main conclusion in last sentence: since the development of the central nervous system of mice and man are not entirely alike, I suggest to be more careful with far-fetched comparisons of the results shown in this manuscript with the clinics of obesity and diabetes. It might be worthwhile adding a short paragraph (or Table) to the Discussion depicting the similarities and differences in neurogenesis and CNS development of humans and mice.
2. Computer-assisted cilia analyses were used throughout this study. While there are a number of references from the authors that describe this in detail, it would be appreciated to mention the method in an abstracted fashion (or schematically).
3. The quality of all images must be improved as the immunofluorescence images are blurry and the charts are very difficult to read.
4. Figure 6 and respective text claim that sex differences are depicted, but that data is not shown in Figure 6. Hence, the Figure caption (line 316) is also misleading. In the legends to Figures 8 and 9, it is stated that datasets were combined when no sex-dependent differences were determined. Perhaps this is also true for Figure 6. Please clarify and/or use the same statement in all captions where it applies to avoid confusion.
Minor points:
5. Some sentences are incomplete or need correction, i.e., sentences in line 109; lines 214-216; lines 443-444; and lines 457-459.
6. Line 173: please specify “GA3 recipe”.
7. Line 225: please explain how a region, or perhaps region-of-interest is meant, can be translated into cilia length.
Reviewer 2 Report
Comments and Suggestions for Authors
Review of Brewer et al.
This paper investigates the ciliary structure (length and presence) in distinct regions of the mouse brain associated with behaviors such as feeding and reward at different stages of embryonic and post-natal development. The two major ciliary markers stained for this evaluation purpose include ARL13B and ADCY3. The authors have used antibodies and reporter fusion protein approach to test the cilia localization. This is an important paper because it provides the first descriptive evidence of presence of cilia in deep brain regions across time that has not been investigated to date in detail. Therefore, the manuscript provides value to the community for testing novel hypothesis associated with ciliary signaling, perhaps in substance abuse research. Some issues that need addressing prior to publication are included below:
Major Issues.
· Historically, cilia length in most tissues are in the 5-10 uM range. The cilia length reported in this work is sometimes greater than 20 uM in length. This calls into question the validity of the quantification method. I would like to see multiple quantitative approaches to ensure that the cilia length reported here is accurate and reliable. The current study relies on AI trained algorithm based in PI’s lab, which will be difficult to replicate for others. A more common source method or previously published and validated cilia software (Lauring et al., Cilia 2019) may provide additional confidence to the data set and increase rigor of these studies.
· In house validation data for the commercial antibodies used to detect the endogenous ciliary proteins must be included. This rigor data will go a long way in convincing readers.
· The rationale for choice of brain regions throughout the manuscript is not done well. Example, in figure 1, SCN brain region IF is depicted and then quants are provided for all regions. In figure 5, ARC brain region IF is depicted and then quants are provided for rest of the brain regions. There is simply no consistency in presentation and the write up similarly does not provide strong rationale for the choices. The authors are requested to go through each panel and their corresponding result section to ensure the flow and logic are clear and easy to follow and understand.
· The figures that were provided were not of high quality and thus difficult to evaluate at times. I don’t know what Figure 1E is showing. I can’t see ciliary structure – just lots of dots in the picture. Not clear the point the authors are trying to make with ISH. Similarly, Fig. 7A is difficult to comprehend.
· Throughout the manuscript, I found numerous sentences that were constructed poorly and did not relay the intended meaning. These phrases need to be corrected. A marked-up MS is provided.
· Quantification presentation in terms of pie charts is easy to view but the percentage of cells should be indicated within the charts for the reader. Figure 4 pie chart data does not match the IF data in the panels. As per the pie chart, ARL13b+ red cells are abundant in all the different brain regions but when I view the IF this cannot be readily seen. This issues goes to the heart of quantification that I mentioned in point 1 above. The quantification must be validated by a second independent approach.
· The title of the MS has two issues – first, it does not accurately reflect the findings because the second ciliary marker ADCY3 was extensively used in all figures. Second, the implication of the term “ARL13B localization” gives the impression of organelle-specific localization differences within neurons, which is not investigated or studied in the manuscript. The location of cilia in distinct brain regions across mammalian development is the main theme of the manuscript and should reflect in the title as well. A title change is recommended.
Minor Issues.
· Please clearly state what control mice are in each panel. If they are non-transgenic WT C57BL/6J mice, please state that instead of “Control” language.
· Presentation – Perhaps create a data in tabular format for easy viewing and comprehension of the data for cilia size, location across various ages. This will help with data interpretation as well.
· Stat test is missing in figure 6 legend. Please check all figure legends for stat test inclusion. Please include ARC and P0 in the IF panels.
· A clear description of embryonic, juvenile, and adult mice ages will help orient the reader to the differences in cilia structure and numbers in each stage in each figure panel.
· Line 329 – include SCN and PVN in the nuclei of hypothalamus sentence.

The English is fine for the most part but there are many poorly constructed phrases that have been highlighted in the attached document provided to the authors.
